# Education for Social Change: The Case of Teacher Education in Wales

Rebecca Weicht [1] and Svanborg R. Jónsdóttir [2,*]

1   Department of Strategy, Enterprise and Sustainability, Manchester Metropolitan University, All Saints, Manchester M15 6BH, UK; r.weicht@mmu.ac.uk
2   School of Education, University of Iceland, Stakkahlíð 1, Reykjavík 105, Iceland
*   Correspondence: svanjons@hi.is

**Abstract:** Entrepreneurial education offers valuable opportunities for teachers to foster and enhance creativity and action competence, which are also important for sustainability education. The University of Wales Trinity Saint David (UWTSD) is a leader in the development of entrepreneurial education in teacher education both in Wales and internationally. The objective of this article is to shed light on how an entrepreneurial education approach can help foster social change. The aim of this study is to learn from teacher educators at UWTSD about how they support creativity, innovation, and an enterprising mindset in their learners. A case study approach is applied. By analysing documentary evidence such as module and assignment handbooks, we explore how teacher educators at UWTSD deliver entrepreneurial education for social change. Our findings indicate that UWTSD's development of entrepreneurial education in teacher training has enabled constructive learning, cultivating creativity and action competence. We provide examples that display how the intentions of the Curriculum for Wales and entrepreneurial education approaches of the UWTSD emerge in practice. These examples show outcomes of the entrepreneurial projects that evince the enactment of social change. The findings also show that the educational policy of Wales supports entrepreneurial education throughout all levels of the educational system.

**Keywords:** entrepreneurial education; sustainability education; social change; creativity; innovation; action competence

## 1. Introduction

Entrepreneurial education has for some time been seen primarily as education about business, with its importance in its potential contribution to the economic progress of societies. Here, we follow the definition of entrepreneurial education as a "catch-all" term that comprises both enterprise and entrepreneurship as outlined in the guidance document "Enterprise and Entrepreneurship Education: Guidance for UK Higher Education Providers" by the UK's quality body for higher education, The Quality Assurance Agency (QAA) [1]. Enterprise education denotes the development of students' capabilities as critical and future-orientated thinkers who are civic-minded and socially responsible. In contrast, entrepreneurship education focuses on fostering the competencies outlined in enterprise education, but within the specific context of creating a new venture [1]. In recent decades, the understanding of entrepreneurial education has widened to encompass an area of learning that cultivates creativity, action, and critical thinking [2–6]. Acknowledging this view, entrepreneurial education can be seen as enhancing personal and cultural growth, economic and technological development, and scientific discovery [7,8]. Some researchers describe the core pedagogy of entrepreneurial education as *emancipatory pedagogy*, where the learners have ample agency and the teacher gradually moves from strong framing towards giving learners total control of their projects [9,10]. To respond to the uncertainties of the future and to imagine possible futures, people in the modern world need the ability to be creative and innovative, as this is important for dealing with the intertwined challenges

of economic, social and environmental issues [11]. The development of *action competence* is also a key to *sustainability education* [12]. Learners of the 21st century need a broad skillset to function in a sustainable world, including collaboration, problem framing, critical thinking, innovation, and creativity [13,14]. Sustainability and education for sustainability are complex endeavours that must build on an understanding of the interconnectedness and multidisciplinarity of the economic, social, institutional, and environmental aspects of society [15–17]. Entrepreneurial education can also drive the changes in education needed to support and inculcate competences for sustainability [5,18].

Yet, while across EU member states (and beyond) there is a consensus for the need for entrepreneurial skills—which are acknowledged to be key to learners' personal and professional lives [19,20]—the teaching of entrepreneurial skills in Europe's schools is patchy. Eurydice [21] found entrepreneurship education in schools to be fragmented and not yet prioritised. Specifically, the researchers found that over half of the researched countries had few or no guidelines on entrepreneurship education teaching methods, and that it was rarely addressed in initial teacher education (but more common in teacher continuous professional development [21]). Equally, no country had fully mainstreamed entrepreneurship education [21].

It is against this background that we can learn from Wales, which has been at the frontier of change in advancing entrepreneurial education both in policy and in practice [22]. The Welsh education sector benefits from over a decade of experience in entrepreneurial teacher training, and Welsh education policy has influenced European-level education policy in relation to entrepreneurial education development [23]. Among other activities, Welsh researchers and educationalists helped develop the European Commission's "Entrepreneurship Competence Framework" (EntreComp) [23], which is the foundation for the EU's Entrepreneurship key competence for lifelong learning [15,20,22,24].

One reason that Wales is advanced in delivering entrepreneurial education is that the Welsh education system has responded swiftly and strategically to repeated criticism of underperformance [25]. In the 2014 Programme for International Assessment (PISA) study, the Organisation for Economic Co-operation and Development (OECD) found that the Welsh education system was producing a high number of "low performers" and that schools were unable to meet learners' needs [26] (p. 7). They also found that inequality persisted because educational results still closely correlated with socio-economic status [25,27]. The OECD thus concluded that the Welsh education system needed a "radical restructuring" [25] (p. 318). Today, the Welsh Government is in the process of rolling out a new curriculum ("Curriculum for Wales", CFW) specifically focussed on skills and on teaching learners to become "ambitious, capable learners, ready to learn throughout their lives" and "enterprising, creative contributors, ready to play a full part in life and work" [28] (p. 11). One of the interesting core concepts in the CFW is *cynefin*, a sense of belonging and identity in a historic, cultural and social context, providing a foundation for a local and international citizenship [28].

However, delivering on such ambitious goals requires enterprising educators [29] such as those at the University of Wales Trinity Saint David (UWTSD). Here, we explore this university as a case study in learning how to help teachers become entrepreneurial educators and successfully deliver the new CFW. Educators and researchers from UWTSD have left a footprint at the international, European, and national levels [30,31]. Among others, they lent their expertise to develop a framework and national teacher training course in North Macedonia in a World Bank-funded programme [21]. Equally, UWTSD's newly developed Doctorate in Education (EdD), which was informed by and integrates the EU's EntreComp framework, has been featured in the "EntreComp into Action" user guide to the framework [32]. At national level, UWTSD professor Andy Penaluna chaired the development of the 2018 updated version of the QAA Guidance for UK Higher Education Providers on Enterprise and Entrepreneurship Education [33].

The purpose of this study is to present an example of educational policy about entrepreneurial education that can be a model for other policy makers in education interested

in sustainability thinking and actions for social change. The aim is to shed light on how Wales has developed entrepreneurial education and how UWTSD has put this policy in practice.

In the following sections, we outline our theoretical framework as derived from Wales' proposal for a "Curriculum for Wales" that is currently being rolled out and seeks to create an entrepreneurial culture. We then present our methods section to finally compare how the core demands of the new curriculum have already been implemented in UWTSD's education, and teacher training specifically, for some time.

## 2. Background

The new Curriculum for Wales (CFW) is a strategic response to criticism on the educational system in Wales. In this section, we first introduce the core elements in the CFW to highlight the innovative aspects in its pedagogy, approaches, assessment, and ideologies. The second part of this section will then outline the theoretical framework that derives from our analysis of the CFW.

### 2.1. Developing the Curriculum for Wales

Wales' new curriculum is the culmination of review and change processes that started over a decade ago. Following a series of bad results in international comparisons, the Welsh government sought to create an entrepreneurial culture both through its curriculum and assessment approaches [22,25]. As early as 2006, a review of initial teacher education in Wales was undertaken and recommendations made on how to improve it [34]. In 2013, as part of a multi-step plan to better standards in Welsh schools, Professor Ralph Tabberer, a former Director-General of Schools in England, also reviewed initial teacher training [35]. Tabberer found the quality of initial teacher training in Wales to be no better than "adequate" and pointed to problems in recruitment, quality, and consistency, as well as a lack of competition among initial teacher training providers in Wales [35] (p. 14). He made 15 recommendations to inform future policy decisions and to raise the quality and consistency of teaching and assessment in initial teacher training, including to improve leadership in the sector and the status of teachers in Welsh society to attract the best candidates to the profession [35].

In 2015, the "Successful Futures; Independent Review of Curriculum and Assessment Arrangements in Wales" report (or Donaldson review) sought to determine how Welsh schools can prepare learners for an uncertain future [33]. Donaldson outlined that education should help learners become enterprising and creative contributors who are ready to play their full part in life and work, as well as ethical and informed citizens who are ready to be citizens of Wales and the world [33]. He focussed his proposal on how to help learners become active citizens and lifelong learners [33]. This played out in a pedagogy focussed on methods to develop skills with an emphasis on progression [33]. In the same manner, assessment should be formative and become an "essential" part of teaching [33] (p. 76). Donaldson's recommendations for how to adapt education in Wales for the future were accepted in full by the Welsh government and became the blueprint for the new CFW, which will be rolled out as of the 2022/23 school year and aims for every child and young person (aged 3 to 16) to become:

1. Ambitious, capable learners, ready to learn throughout their lives;
2. Enterprising, creative contributors, ready to play a full part in life and work;
3. Ethical, informed citizens of Wales and the world;
4. Healthy, confident individuals, ready to lead fulfilling lives as valued members of society [28] (p. 11).

The CFW has been described as a "bold new vision for curriculum, teaching and learning" and a "radical departure from the top-down, teacher proof policy of the previous National Curriculum" [36] (p. 181–182). Others noted that "pupils will encounter knowledge very differently from previous generations" because of its move away from subjects and the autonomy it offers schools and teachers on how to deliver content, as well as its

learner-centred focus and focus on transversal skills [37] (p. 7). Power et al. report that teachers are both excited and nervous about the new curriculum [24]. They are excited because they see the new curriculum as less "prescriptive" and thus suffering less from burdensome administrative work [38] (p. 5), but also nervous in the face of the unknown.

The CFW outlines the details of the new curriculum's aims and objectives [28]. It explains that the aim for learners to be ambitious and capable means that they are both capable of solving problems, and also enjoy the challenge of solving them. To be enterprising, creative contributors requires learners to be creative in their approach to solving problems and to "give of their energy and skills so that other people will benefit" [28] (p. 24). To be an ethical, informed citizen means to consider others, the environment, and one's actions, and to be ready to engage with contemporary issues based on one's values [28].

Specific skills are considered "integral" to enabling the four purposes of the CFW [28] (p. 25). They are 1. Creativity and innovation, 2. Critical thinking and problem-solving, 3. Personal effectiveness, and 4. Planning and organising. Creativity and innovation mean for learners to be curious and inquisitive. Personal effectiveness means that learners are able to evaluate their thinking and mistakes and to be able to identify and recognise different types of value [28]. Planning and organising is the ability to put ideas into action. The guidance specifically states:

When developing these skills, learners should:

- Develop an appreciation of sustainable development and the challenges facing humanity;
- Be afforded the space to generate creative ideas and to critically evaluate alternatives—in an ever-changing world, flexibility and the ability to develop more ideas will enable learners to consider a wider range of alternative solutions when things change [28] (p. 26).

The curriculum clearly calls for empowering learners to become active agents of building a socially just and sustainable society. The guidance asks learners to "appreciate the contribution they and others can make within their different communities and to develop and explore their responses to local, national and global matters" [28] (p. 30).

Until the CFW is rolled out across Wales, so-called "Pioneer Schools" are tasked with the development of the curriculum in more detail. The CFW guidance document so far only outlines six "areas": Expressive Arts, Health and Well-being, Humanities; Language, Literacy and Communication; Mathematics and Numeracy; and Science and Technology. It does not break down knowledge into subjects that should be taught at different levels [28]. Instead, the focus is on interdisciplinary, and student-centred, active learning with real-life relevance [24]. The Welsh government believes that schools are best placed to make decisions about learners' needs and has thus tasked schools in their detailed guidance to develop the curriculum based on "What matters" statements [24]. Upon implementation in schools, it is then up to the schools to develop a vision for themselves and design a curriculum to implement that vision in their school [28].

### 2.2. What the CFW Means for Teacher Agency and Expectations

Such an innovative approach to curriculum and teaching demands a lot of autonomy and agency from teachers [28]. It is here that teacher education that focuses on developing innovation and action skills of teachers becomes crucial. Newton et al., in surveying teachers and headteachers, found that teacher perception of the new curriculum depends on their schools [38]. They found that, while the perception of the new curriculum is shaded by the "bad" perception of the current curriculum, respondents felt that the Pioneer Schools had a more positive outlook than schools outside the Pioneer School network [38] (p. 9). This may be because the Pioneer Schools are better prepared and have access to more and better resources. Their survey respondents also described the Pioneer Schools as "innovative" and "progressive" places where teachers already used the teaching methods as laid out in the Successful Futures report, the foundation for the CFW [38]. They quote one teacher as stating:

There's nothing really new in Donaldson it's just good teaching, and the good teachers have been teaching in an Donaldson-esque way for a considerable length of time, it's just they didn't know what it was. It's just good teaching—making sure that it's relevant to the pupils  [38] (p. 39).

Teachers who positively anticipate the new curriculum expect to see benefits from the focus on progression over attainment [38]. They expect that this approach allows for greater recognition of the different ways of learning achievements. For example, the focus on formative assessment is expected to be "more likely to support multiple pathways to learning" [38] (p. 13). Equally, the autonomy to move away from the previous model of tests and exams is viewed positively [25].

Therefore, the importance of teacher training in (successfully) delivering the new curriculum lies in giving teachers the tools and fostering the mindset to try new ideas and use the space they are given. It is here that the importance of innovative teacher training comes in. Teacher trainers and educators alike need to share good practices to help teachers adapt to the mindset shift that the CFW demands. In the words of Welsh educators:

"It's a difficult one because it's 'change your mindset' more than resource."

"I think a lot of heads will need to become far more creative and change their mindsets, look at the curriculum design issue. It's not going to be a box ticking exercise thank god, we've had that. This has got to be a lot more evolved and it's got to be a change of mindset."

"Have they got the skills to do those things because we've never taught in that particular way and you can't just suddenly change the mindset of a profession that's almost going to take a generation to re-educate that profession to do things differently"  [38] (p. 47).

### 2.3. Curriculum for Wales Delivery and Assessment

The following section outlines in detail how education delivery and assessment are described in the CFW, as well as where there is an overlap with sustainability education and education for social change. The understanding of how the CFW is meant to be delivered, how assessment is meant to take place, and how the CFW views active and responsible citizens will then guide the empirical analysis in the following sections [39].

#### 2.3.1. CFW Delivery

The CFW specifies that the delivery of the curriculum should make use of external practitioners and their expertise. For example, in the Expressive Arts Area of Learning and Experience, this may include visits to theatres and galleries as well as bringing the expertise of external practitioners into the classroom [28]. Relatedly, to enhance learners' skills, learning and teaching should be delivered using a range of teaching and learning approaches, including digital ones. For instance, in the Mathematics and Numeracy Area of Learning and Experience, learners may work together using digital skills and to solve a problem and develop an algorithm that supports the understanding of patterns. They may also use digital tools to create graphs from spreadsheets, for example [28].

Similarly, learning and teaching should take place in a range of contexts and be cross-curricular. The CFW is cross-disciplinary within and across Areas. It also sets out three mandatory cross-curricular skills: literacy, numeracy, and digital competence [28] (p. 13). The mandatory cross-curricular skills "are essential to all learning and the ability to unlock knowledge. They enable learners to access the breadth of a school's curriculum and the wealth of opportunities it offers, equipping them with the lifelong skills to realise the four purposes" [28] (p. 27). For each of the six thematic areas, the mandatory cross-curricular skills are mapped out. For example, in the Well-being Area of Learning and Experience, numeracy "is a key enabler in making a number of informed decisions, in particular managing money and supporting good financial decision-making and critically engaging

with social norms around money. Numeracy also plays a role in purchasing and preparing food to support nutrition" [28] (p. 85).

Learning and teaching should also ensure exposure to local, national, and international contexts at different stages of development. In addition, learning and teaching should take place in authentic contexts. For example, collaboration with individuals and employers provides learners with opportunities to learn about work, employment, and the skills valued in the workplace [28] (p. 44). This may also lead them to develop enterprise activities, which can provide authentic learning experiences that contribute to their development as enterprising, creative contributors to society. Equally, engaging parents and caregivers, school partners, and the local community can create authentic, contextualised learning opportunities. For instance, the overlap between the Humanities Area and the Mathematics and Numeracy Area might include the collection of a range of qualitative and quantitative primary data [28].

Furthermore, learning and teaching should allow for learners to develop their skills (e.g., critical thinking, problem solving, and decision-making) and for them to generate different types of value (financial, cultural, social, and learning) [28]. The integral skill "creativity and innovation" should support learners in creating different types of value.

Lastly, learning and teaching may be based on a whole-school approach. For example, in the Languages, Literacy and Communication Area, the CFW refers to the Content and Language Integrated Learning (CLIL) approach. It is stated that effective language learning requires a "systematic whole-school approach" that requires that schools be "aware how best to ensure progression for all learners in all their languages, for example through immersion, Content and Language Integrated Learning (CLIL) or plurilingual activities" [28] (p. 160).

All in all, the delivery of the CFW appears to be tailored as a holistic design that emphasises the interconnectedness of the elements within the whole, and where each area of learning, skill, or competence engages learners in a meaningful way.

### 2.3.2. Assessment in the CFW

Assessment in the CFW is formative and progression-focussed [28]. It should be ongoing to help the learner identify their strengths and improve their weaker areas. It should guide the learner to the steps needed to progress. The "overarching purpose is to support" and move learning forward [28] (p. 9). Progress is measured based on the statements of what matters [28]. The additional principles of progression aim to give educators a better understanding of progression [28]. They apply across the curriculum and explain what progress may look like and which principles underpin progress. These principles are: increasing breadth and depth of knowledge, deepening understanding of the ideas and disciplines within the Areas, refinement and growing sophistication in the use and application of skills, making connections and transferring learning into new contexts, and increasing effectiveness [28] (p. 129–131). In short, evaluation and improvement through reflection are core to the new curriculum and make "a vital contribution to raising the quality of education and standards of achievement" [28] (p. 229).

Assessment should also be holistic in providing insights into the learner's learning needs. It should include a wide range of assessment approaches to provide a full picture of the learner's development. It should include assessments by educators as well as the learner themselves. This may take place, for example, via portfolios that allow the learner to visualise their progress over time. Assessment should occur not just in the school but in exchange with the wider world as well. There should be engagement between the learner and the world around them, including parents or caregivers and practitioners. For instance, in relation to learning about careers and work-related experiences, the CFW indicates that experiences should stimulate an interest in different careers and that learning should take place in practical ways. Entrepreneurial activities, for example, necessitate reflection as a learning skill. They relate to practical activities such as business start-ups or venture-creation programmes [28] (p. 43). Similarly, in the arts, self-evaluation and

reflection are part of the integral skills of critical thinking and problem-solving. The CFW explains:

> Refining work is encouraged throughout one of the statements of what matters in this Area, with the aim of building skills in self-evaluation and reflection. The evaluation involved in the creative process enables learners to develop reflective, questioning and problem-solving skills, as well as to challenge perceptions and identify solutions. Learners may demonstrate resilience in applying critical appraisal of their work and be expected to respond positively to critical feedback. Learners can develop problem-solving skills by experimenting with a variety of arts and artistic techniques [28] (p. 65).

### 2.3.3. Active, Informed, and Responsible Citizens

Many of the elements of the CFW indicate a will to inculcate in learners a competence for action in developing a socially just and sustainable society. Learners are meant to engage with important issues facing humanity, such as sustainability and social change, and to develop the skills necessary to do so. They are expected to learn to become active, informed, and responsible citizens and consumers who can identify with and contribute to their communities and reflect on the impacts of their actions. As in entrepreneurial education, there is an emphasis on a connection with society, authentic contexts, and cross-curricular areas through real life experiences. Learners are meant to learn to exercise their democratic rights, imagine possible futures, and take social action. They are expected to know or participate in enterprise and entrepreneurial activities and social action projects [28] (p. 123). Learners' creativity is meant to be stimulated and their capacity to produce solutions should grow as they engage with ethical issues of sustainability and business [28]. They must be able to make responsible decisions when acting socially, politically, economically, and entrepreneurially [28] (p. 102).

The CFW suggests that learners can get to know and explore the multiple and connected issues of sustainability through entrepreneurial education. Entrepreneurial education affords them opportunities to "understand the interconnected nature of economic, environmental and social sustainability; justice and authority; and the need to live in and contribute to a fair and inclusive society" [28] (p. 102) as learners get to experience real-life enterprises applying their own creativity and action competence in collaboration with others. A fascinating concept is presented in the CFW, *cynefin*, taken from Welsh:

> The place where we feel we belong, where the people and landscape around us are familiar, and the sights and sounds are reassuringly recognisable. Though often translated as 'habitat', cynefin is not just a place in a physical or geographical sense: it is the historic, cultural and social place which has shaped and continues to shape the community which inhabits it" [28] (p. 241).

Cynefin seems to embrace the individual, the local and global in understanding oneself as a part of a community and culture and realising how choices we all make can have impacts on society.

Tables 1–3 summarise the major themes of how the CFW should be delivered, assessed, and where it overlaps with education for social change and sustainability through entrepreneurial education. We will use these tables to identify how and where the characteristics from the CFW emerge in UWTSD documents.

**Table 1.** Learning and teaching delivery as outlined in the CFW.

| Learning and Teaching in the CFW |
|---|
| • Learning and teaching should be collaborative and cross-disciplinary (pp. 6, 47, 50) |
| • Learning and teaching should make use of external practitioners (e.g., pp. 53, 88, 227–228) |
| • Learning and teaching should be delivered using a range of teaching and learning approaches (e.g., pp. 35, 57, 89, 116, 119) |
| • Learning and teaching should take place in a range of contexts and be cross-curricular (e.g., pp. 8, 24, 26, 34, 44) |
| • Learning and teaching should allow for learners to develop their skills (e.g., critical thinking, problem solving, decision-making) and for them to generate different types of value (financial, cultural, social, learning value; e.g., pp. 6, 23–26) |
| • Learning and teaching should ensure exposure to local, national and international contexts at different stages of development (e.g., pp. 30, 102) |
| • Learning and teaching should take place in authentic context(s) (e.g., pp. 66, 96) |

**Table 2.** Assessment as outlined in the CFW.

| Assessment in the CFW |
|---|
| • Assessment should enable reflection on learner progress over time (e.g., it should inform a learner on their strengths and achievements, as well as areas for improvement and, if relevant, barriers to learning; e.g., p. 8) |
| • Assessment should also enable reflection on group progress over time (e.g., at school level) |
| • A wide range of assessment approaches should be used to provide a holistic picture of learners' development (pp. 6, 31) |
| • There should be engagement between the learner and the world outside of school, incl. parents or carers, and practitioners (p. 226) |
| • Learners should participate in the assessment process (e.g., reflect on their learning journey; e.g., pp. 51, 92, 157, 185) |
| • As learners progress, they should become increasingly effective. This includes increasingly successful approaches to self-evaluation, the identification of their next steps in learning and more effective means of self-regulation (p. 30) |

**Table 3.** Education for social change as outlined in the CFW.

| Education for Social Change in the CFW |
|---|
| • Learners should be empowered to become active agents of building a socially just and sustainable society (e.g., pp. 12, 19, 30, 41, 42, 76, 97, 98, 102) |
| • Learner engagement is emphasised. Learning should take place in authentic contexts across curricular areas (pp. 48–50) |
| • Learners should adopt an enterprising spirit and action competence. Learners should be able to create value of different kinds—financial, cultural, and social (p. 25) |
| • Learners should become enterprising in managing their own and others' resources, valuing failure as a part of the creative process, and relatedly strengthening their employability skills (e.g., pp. 65, 73, 85, 98, 123) |
| • Learners should become sustainable citizens through a sustainable education, and should be able to respond to challenges of a social, economic and environmental nature (e.g., pp. 45, 70, 98, 102, 120) |
| • Learners should be able to make responsible decisions, to act as caring, participative citizens of their local, national, and global communities, committed to justice, diversity and the protection of the environment (e.g., pp. 70, 98, 102) |

## 3. Methods

We apply a case study methodology to achieve our research objective. With its focus on understanding the how and why of a social phenomenon in context, a case study lends itself to learning from Wales, and UWTSD in particular, to understand how educators are supported to cultivate competences for social change in learners [40].

To address our research aim, we drew on more than a dozen data sources. We drew on peer-reviewed papers, internal and external UWTSD documents such as Annual Reports as well as programme handbooks, and websites or blogs (see Table 4). For instance, where they were available, we sought to primarily rely on peer-reviewed papers to highlight the entrepreneurial education approach used by UWTSD (e.g., [37,41]). This means that our data sources had already passed a level of quality control through peer-review. We considered it appropriate to draw on websites such as project websites and, for example, their blogs (e.g., [42,43]) to illustrate how some of these concepts are applied in practice. All data are listed in the reference list. Most documents are available online and can be accessed freely. All accessed materials were in English. By triangulating these different forms of data (academic papers, blog posts, project and annual reports, websites), we seek for our case study to become a rich and robust account that is comprehensive and well developed, thus helping facilitate deep understanding [44]. Thick description then allows us to evaluate to what extent our conclusions may be transferable [44] and to illustrate the theory and how we came to assess if and that UWTSD met the CFW spirit.

**Table 4.** Overview of data sources.

| Document Type | Number |
| --- | --- |
| Academic papers (incl. conference presentations) | 8 |
| Reports (e.g., technical policy reports) | 3 |
| External UWTSD documentation | 1 |
| Internal UWTSD documentation | 2 |
| Other resources (e.g., project websites and blogs) | 2 |

The authors started collaborating on the research project in early January 2021 by designing the research and dividing work. We held on-line hour-long meetings every 10–14 days where we discussed the process and reflected on and responded to what was emerging. The data collection and analysis was conducted iteratively as we first scanned major documents (such as the CFW and module handbooks) looking for signs of answers to our research questions. As we started writing up, we consulted the documents and added others that helped to achieve a clearer picture (e.g., blogs, reports, and academic publications). We reported regularly to each other what results we were producing and asked the other to reflect and comment on emerging findings. We kept notes of those meetings to go back to where necessary. Furthermore, we benefitted from feedback from UWTSD researchers and authors of some of the papers we analysed where we encountered issues.

Both authors have experience of teaching entrepreneurship education and have taken part in European projects in collaboration with teachers from UWTSD. The second author specialises in entrepreneurial education and has worked on research for the last decade in that area, mainly with qualitative methods. The first author has contributed to publications on entrepreneurial education for, among others, the European Union and published in the field. Our views are positive towards entrepreneurial education and to the quality of the work we got to know constructed by UWSTD in this area. We were aware of our attitudes and regularly reminded ourselves of the impact they might have on our results.

Both researchers involved in this research project have in the past collaborated with UWTSD entrepreneurial education researchers and educators on different projects. We were thus able to draw upon our experience and knowledge in terms of publications that outline the UWTSD approach. We were also able to identify where projects (those we participated in as well as others) were useful illustrations of different educational theories, e.g., "glorious failure" [45]. Additionally, we benefitted from support from personal contacts to UWTSD researchers to point out to us other publications and projects that could be of value. This support allowed us to achieve data saturation as we were able to find high-quality examples in the public domain that could be of interest to illustrate how UWTSD supports education for social change along the understanding of the CFW.

## 4. Results

The following sections will show how UWTSD already implements the dictates of the CFW in its teacher training and wider education approach. Examples are provided to highlight good practice(s) of an entrepreneurial education approach for social change and sustainability.

### 4.1. Delivery

Several core aspects of learning and teaching at UWTSD reflect various tenets of the new CFW delivery.

First, the CFW integral skill "creativity and innovation" is deeply rooted in UWTSD's entrepreneurial education. UWTSD has a long history of drawing on design education for entrepreneurial education [46–50]. Its learning and teaching approaches borrow methodologies from design to create value for others through seeing multiple perspectives. As early as 2008, researchers at the International Institute for Creative Entrepreneurial Development (IICED) at UWTSD proposed "curiosity-based learning" [46] as a strategy for learners to recognise problems and generate solutions. As the authors wrote:

> Reflecting the inquisitive entrepreneur, learners become aware of their shortfall in knowledge through their own experience, rather than simply being told it. They also learn to look around a problem and not just to see it at face value, or are encouraged to find problems within scenarios presented to them as a project assignment brief. Finding these problems is a necessity—failure to do so results in learners not being able to engage with the scenario [46] (p. 405).

UWTSD has taken this focus on creativity to an international forum, having contributed to United Nations education programmes, among others. Together with the United Nations Conference on Trade and Development, IICED developed a curriculum that aimed to enhance innovative capacity among learners in educational programmes [30]. For the OECD, their researchers authored a "Thematic Paper on Entrepreneurial Education in Practice" arguing that "retaining the creative thinking of the young mind is important and real world relevance and levels of connectivity will help to bring invaluable insights to our schools" [29] (p. 7).

Relatedly, real-world context plays an important role in UWTSD's teaching and learning, which is collaborative both within and beyond the classroom. UWTSD has a long history of engaging alumni as sources of information for evaluation and as external practitioners [28]. Within the classroom, learners engage in project work to develop their knowledge and ideas with both peers and teaching staff [47]. At the same time, engagement with these speakers ensures that learning and teaching take place in authentic contexts. As part of the Postgraduate Certificate in Education (PGCE), Professional Certificate in Education (PCE), and Professional Certificate in Education for Post Compulsory Education and Training (PCET) programmes, guest speakers (including students in the Education Doctorate programme, EdD) are invited to inform trainee teachers of recent developments within the field of post compulsory education and to enable learners to view education in relation to different contexts. Among others, Wales has an established method of providing Entrepreneurship Champions, who before they join any classes have to attend a short course as an introduction to learning and teaching.

Equally, learning and teaching are focussed on skills development for learners (including trainee teachers) to appreciate that value creation can go beyond economic value.

Table 5 outlines in detail the CFW requirements in relation to delivery, shows how UWTSD implements them, and provides examples.

**Table 5.** Delivery as outlined in the CFW guidance mapped against UWTSD implementation and visualised by examples.

| CFW Guidance on Delivery | UWTSD Implementation | Example |
|---|---|---|
| Learning and teaching should be collaborative and cross-disciplinary | Trainee teachers at UWTSD are engaged in collaborative teaching and peer support to extend the range of strategies and methods they employ within their teaching and to seek to continually improve themselves to the benefit of their learners. | As part of the Professional Graduate Certificate in Education, trainee teachers are engaged in workshops to practice and develop their skills in teaching, research and critical analysis, resource development, experimentation with traditional and creative pedagogies, application of digital technology skills, self-reflection, and peer and self-evaluation; they also engage in project work to develop their knowledge and ideas with both peers and teaching staff [50]. |
| Learning and teaching should make use of external practitioners | UWTSD has a long history of engaging alumni as sources of information for evaluation and as external practitioners. | UWTSD alumni have contributed to programme development. Especially, their "ideas, perceptions and experiences, networks and contacts, have provided particularly rich empirical evidence that enabled comprehensive and detailed consideration and evaluation" [37] (p. 234). |
| Learning and teaching should be delivered using a range of teaching and learning approaches | UWTSD designs "fit for purpose" learning and assessment, ensuring that is it constructively aligned. Cognition research into insightful as well as analytical thinking theoretically underpins this approach. | Penaluna et al. [46] explored practical measures of how student performance can be assessed and argued that inappropriate assessment strategies can significantly inhibit the creativity of students and teachers. |
| Learning and teaching should take place in a range of contexts and be cross-curricular | UWTSD's approaches borrow learning methodologies from design education, which seeks to create value for others through seeing multiple perspectives within wicked problem-solving scenarios. | For example, UWTSD makes use of pedagogies such as curiosity-based learning that distinguishes between, at first, focussing on divergent thinking (opening minds and synthesis), and then converging (analytical and solution-focused) to generate ideas and explore possible solutions [47–49]. |
| Learning and teaching should allow for learners to develop their skills (e.g., critical thinking, problem solving, decision-making) and for them to generate different types of value (financial, cultural, social, and learning) | For over 10 years, UWTSD has been involved in various international innovation projects focussed on mainstreaming entrepreneurial education skills among teachers. | UWTSD was involved in the EU-level ADEPTT (http://adeptt.blogspot.com/, accessed on 4 June 2021), Eco System App (https://ecosystemapp.net/, accessed on 4 June 2021), EntreAssess (http://entreassess.com/, accessed on 4 June 2021), and EntreCompEdu (https://entrecompedu.eu/, accessed on 4 June 2021) projects. Dr. Jan Barnes, UWTSD Senior Lecturer in Cross-curriculum close to practice enquiry and research, described for the EntreAssess project in 2018 how she develops trainee teachers' entrepreneurial skills [42]. The EU-funded policy reform project EntreCompEdu, led by UWTSD, developed a professional skills framework of entrepreneurial education and ran a teacher training course with over 400 teachers globally [41]. |
| Learning and teaching should ensure exposure to local, national, and international contexts at different stages of development | UWTSD has been involved in international teaching and research projects to develop teacher education for many years. Learners participate regularly in these opportunities. | Educators engaged in the development of the EntreCompEdu programme. As part of this training programme, they engage with educators from other countries, learn together, and exchange teaching ideas [41]. UWTSD also co-hosts international events, e.g., as part of Global Entrepreneurship Week, for their students [51]. For example, they hosted Fiorina Mugione, previously the Chief of Entrepreneurship at the United Nations. |
| Learning and teaching should take place in authentic context(s) | UWTSD makes extensive use of guest speakers (including alumni) in their teacher training. | For instance, guest speakers are invited to inform trainee teachers of recent developments within the field of post-compulsory education and to enable learners to view education in relation to different contexts. This also includes EdD students. See also this blog reflecting on an alumni-led event and [37,49] (https://www.lancaster.ac.uk/users/enterprisecentre/stepping-back-to-go-forward-alumni-voices/, accessed on 31 May 2021) |

*4.2. Assessment*

In relation to assessment, UWTSD supports students extensively in reflecting on their learning progress over time. This is core to the arts and design education that much of UWTSD's work is built upon [29,46,50,51]. Students are evaluated through summative and formative assessment methods such as project work, presentations/pitches, self-reflection, as well as self-evaluation, peer evaluation, and external expert review/feedback [45]. They are encouraged to "fail fast" and to make mistakes and learn from them ("glorious failures") [29,45]. For instance, students take part in "The Crit", an arts-based discussion where students critique each other's work. Penaluna and Penaluna described this as follows:

> Students are expected to communicate and debate their thinking processes and enter into a discussion of their work with tutors and peers and in later studies when appropriate with external stakeholders such as industry practitioners, clients and community members [49] (p. 7).

The approach forces students to explain their decision-making in envisioning new futures as well as to build on feedback [49]. Across their learning, progress is mapped by charts and evidence that showcase which connections the student has made in their mind. The more complex and numerous these connections are, the better:

> The highest grades are given to those who can argue for a range of distinctly different yet justifiable solutions. The number of alternative solutions required will be determined by the educator, who will consider the developmental stage of the learners. New students may be asked to present only two alternatives, whereas more accomplished students will be more challenged, with 6 to 12 alternatives [49] (p. 7).

In addition, UWTSD has become adept at involving alumni in assessment approaches based on the long-standing "Continuous Conceptual Review" model [47]. Among other forms of participation, alumni join classes to reflect on their learning in their own contexts.

UWTSD is also involved in research and policy projects to foster entrepreneurial education assessment approaches. UWTSD educators have been part of the EU-funded "EntreAssess" project, which published assessment methods, tools, and examples to help educators assess entrepreneurial teaching and learning [52]. This project collated entrepreneurial education assessment methods, tools, and examples from across the world to develop a self-assessment tool and model to help educators understand their own assessment approaches and grow in their use of more creative and complex methods and tools. The different stages range all the way up to a whole-school approach. On the project blog, UWTSD educators provided insights from their own classrooms and how they encourage creative assessment approaches. For instance, Tom Cox, UWTSD Senior Lecturer in Creative and Innovative Teaching and Learning, outlined how he uses the EntreAssess tools in helping primary school teachers to find specific assessment methods for specific skills development [52]. Case studies of Welsh schools are also provided. Craigfelen Primary School in Swansea was among the first schools in Wales that participated in UWTSD's teacher training programmes. In a blog post on the EntreAssess website, Andy Penaluna narrated a visit he undertook to Craigfelen following an invitation to their year 1 and 2 learners, explaining how the learners (5–7 years old) ran a project to revitalise the local post office and summarising the learnings he drew from the visit. The learners opted to run a pop-up shop in the space and involved their parents as well as the local community and UWTSD in this project.

Table 6 outlines in detail the CFW requirements in relation to assessment, shows how UWTSD implements them, and provides examples.

**Table 6.** The CFW requirements on assessment mapped against UWTSD implementation and visualised by examples.

| CFW Guidance on Assessment | UWTSD Implementation | Example |
|---|---|---|
| Assessment should enable reflection on learner progress over time (e.g., it should inform a learner on their strengths and achievements, as well as areas for improvement and, if relevant, barriers to learning) | UWTSD supports learners extensively via formative feedback. | An adaptation of the Art and Design "Crit" is used to enhance peer to peer learning—through the justification of multiple solutions, which are expected to be argued for their distinctiveness from each other (Forced Divergent Thinking) [37,49]. |
| Assessment should enable reflection on group progress over time too (e.g., at school level) | Alumni join classes to reflect on their learning in their own teaching contexts. | "Glorious Failure" is a teaching/assessment approach in which students are allowed to "fail" if they reflect upon the why and articulate the reasoning [45]. |
| A wide range of assessment approaches should be used to provide a holistic picture of learners' development | Students are subject to various summative and formative assessment methods such as project work, presentations/pitches, self-reflection, as well as self-evaluation, peer evaluation, and external expert review/feedback. In addition, UWTSD has been and is actively involved in research in entrepreneurial education assessment to help educators develop progress in their assessment challenge. | UWTSD educators have been part of the EU-funded EntreAssess project that published assessment methods, tools and examples to help educators with assessing entrepreneurial teaching and learning. The project focussed on practical and easy-to-use assessment methods and aimed to help enhance students' learning in entrepreneurial education and support the quality of education and outcomes in European contexts [52]. |
| There should be engagement between the learner and the world outside of school, incl. parents or carers, and practitioners | UWTSD has a long history of engaging alumni as sources of information for evaluation and as external practitioners. | If an educator is innovative, they can "only be realistically evaluated and validated by their learners", in other words: alumni [45] (p. 29). Consequently, alumni engagement and the empowerment of alumni to return to university and provide advice as well as share their experience of their education, for instance, preparing them for their work, are crucial. See also [47]. |
| Learners should participate in the assessment process (e.g., reflect on their learning journey) | Students are encouraged to "fail fast" and to be comfortable making mistakes and learning from them ("glorious failures"). | "Glorious failures" denotes a concept that accepts that what is new will likely be a prototype that is improved with testing and feedback. In education, it means understanding and accepting that interventions will be prototypes, which means for both students and educators, the experience itself will be as valid as the immediate outcome [45]. |
| As learners progress, they should become increasingly effective. This includes increasingly successful approaches to self-evaluation, identification of their next steps in learning and more effective means of self-regulation | UWTSD is continuously involved in developing training programmes focussed on teachers' continuous development. | UWTSD has been the project lead for the EU-funded policy reform project EntreCompEdu (2018–2020). It supports educators in developing their entrepreneurial education skills. The EntreCompEdu framework builds on good pedagogy in the field of entrepreneurial education. The framework rests on six pedagogical principles: (i) think creatively, (ii) look to the real world for inspiration, (iii), promote collaboration with a purpose, (iv) create something of value for others, (v) stimulate reflection, flexible thinking and learning from experience, and (vi) make entrepreneurial learning visible [41]. |

### 4.3. Social Change, Sustainability, and Entrepreneurial Education

The University's sustainability statement commits UWTSD to deliver "meaningful and relevant educational pathways". This includes promoting learning and social responsibility, which supports what the Brundtland Commission in 1987 described as "development that meets the needs of the present without compromising the ability of future generations to meet their own needs" [50]. UWTSD thus empowers learners to become active agents in building a socially just and sustainable society. In detail, the PGCE/PCE programmes aim to produce learners who "understand their professional responsibilities in relation to Education for Sustainable Development (ESD), with particular regard to the development of practice and engagement within the classroom and the ability to understand, critically evaluate, and adopt thoughtful sustainability values." Teacher trainees are encouraged to be experimental in their teaching and "experiment with pedagogies that embed ESD and consider sustainability through critical reflective practice and evaluation" [50] (p. 31).

Learning is delivered in authentic contexts. For UWTSD, collaboration with industry is a key focus. This facilitates "more value creation opportunities for students" while also augmenting learners' employability prospects [51]. At the same time, industry collaboration allows UWTSD learners to explore value creation opportunities for the development of new sustainable businesses, products, and services. In 2018–2019, UWTSD was ranked first in Wales and second in the UK by the Higher Education Statistics Agency (HESA) for the number of graduate businesses running for three years or more [51]. Over 550 alumni are enhancing and supporting UWTSD's entrepreneurial education ambitions. This demonstrates how UWTSD, in teaching about (and through) entrepreneurial education, focuses on value creation that is ecological, humane, and social, in addition to creating economic value. In 2020, UWTSD teaching staff launched the "Harmonious Entrepreneurship Society" to "set up and advance harmonious approaches to entrepreneurship to address the sustainability challenge facing our planet" [51].

Table 7 outlines in detail the CFW expectations in relation to social change and sustainability, shows how UWTSD implements them through entrepreneurial education, and provides examples.

**Table 7.** The CFW expectations on social change mapped against UWTSD implementation and visualised by examples.

| CFW Guidance on Social Change | UWTSD Implementation | Example |
|---|---|---|
| Learners should be empowered to become active agents of building a socially just and sustainable society | The University's sustainability statement commits UWTSD to deliver "meaningful and relevant educational pathways." This includes promoting learning and social responsibility, which supports what the Brundtland Commission in 1987 has described as "development that meets the needs of the present without compromising the ability of future generations to meet their own needs" [53]. | The PGCE/PCE programmes aim to produce learners who "understand their professional responsibilities in relation to ESD, with particular regard to the development of practice and engagement within the classroom, and the ability to understand, critically evaluate and adopt thoughtful sustainability values." Teacher trainees are encouraged to "experiment with pedagogies that embed ESD and consider sustainability through critical reflective practice and evaluation" [50] (p. 31). |
| Learner engagement is emphasised. Learning should take place in authentic contexts across curricular areas | For UWTSD, collaboration with industry is a key focus. It facilitates "more value creation opportunities for students" while also augmenting learners' employability [53]. At the same time, industry collaboration allows UWTSD learners to explore value creation opportunities for the development of new sustainable businesses, products, and services. | UWTSD has been ranked 1st in Wales and 2nd in the UK in 2018/19 by the HESA for the number of graduate businesses running for three years or more [51]. Over 550 alumni are enhancing and supporting UWTSD's entrepreneurial education ambitions. |
| Enterprising spirit and action competence. Being able to create value of different kinds—financial, cultural and social | UWTSD in teaching about (and through) entrepreneurial education focuses on value creation that is ecological, humane and social, in addition to the traditional economic value creation. | In 2020, UWTSD teaching staff launched the "Harmonious Entrepreneurship Society" to "set up and advance harmonious approaches to entrepreneurship to address the sustainability challenge facing our planet" [51]. All units in the PGCE/PCET/PCE programmes are also mapped against the university's "Education for Sustainable Development" plan, which outlines skills developed in the teacher trainees [50]. |

**Table 7.** *Cont.*

| CFW Guidance on Social Change | UWTSD Implementation | Example |
|---|---|---|
| Learners should become enterprising in managing their own and others' resources, valuing failure as a part of the creative process, and relatedly strengthening their employability skills | Creativity and innovation are at the heart of UWTSD's mission to enhance graduate employability and the number of graduate start-ups. | UWTSD has been ranked 1st in Wales and 2nd in the UK in 2018/19 by the HESA for the number of graduate businesses running for three years or more [51]. Over 550 alumni are enhancing and supporting UWTSD's entrepreneurial education ambitions. |
| Learners should become sustainable citizens through a sustainable education, and should be able to respond to challenges of a social, economic and environmental nature | UWTSD's Sustainability Statement focuses on providing meaningful education that considers social responsibility and the needs of future generations. | The university aims to "utilise our collective skills, knowledge and technology to enable the University and its graduates to offer solutions to the most urgent societal challenges—in Wales and further afield. We are also committed to building a sustainable society driven through enterprising innovation and entrepreneurship" [51] (p. 3). |
| Learners should be able to make responsible decisions, to act as caring, participative citizens of their local, national, and global communities, committed to justice, diversity and the protection of the environment | Entrepreneurial value creation—where value may be cultural, social, or environmental, in addition to economic—is well understood in UWTSD teaching. | UWTSD's "Harmonious Entrepreneurship Society" was set up to advance entrepreneurial approaches that have sustainability at their heart [51]. |

## 5. Discussion

This study sought to present how a university works towards embedding sustainability thinking and actions for social change through an entrepreneurial education approach. It analysed the very forward-looking curriculum demands that Wales makes towards its educators in the "Curriculum for Wales", and mapped against these demands how a specific university with a track record in delivering entrepreneurial education enables education for sustainability and social change. The results highlight some noteworthy practices. Through some of its core entrepreneurial education approaches, UWTSD manages to seamlessly embed sustainable thinking and education for social action in their learning and teaching.

First, through their long history of making use of external practitioners, UWTSD is able to foster an enterprising spirit and action competence in learners. Learners are then able to create value of different kinds. UWTSD, in teaching about—and importantly, through—entrepreneurial education allows broad thinking about value creation that goes beyond the creation of economic value and is ecological, humane, and social. For instance, for UWTSD, collaboration with industry is a key focus. It facilitates "more value creation opportunities for students" while also augmenting students' employability prospects [53]. At the same time, industry collaboration allows UWTSD students to explore value creation opportunities for the development of new sustainable businesses, products, and services. Equally, trainee teachers at UWTSD engage with guest speakers, including alumni, who become participants in UWTSD's teaching approach. Specifically in teacher education, UWTSD includes in-house (budding) experts such as EdD students to inform trainee teachers of recent developments in the field of education. Such approaches aim for learners to be able to view education in relation to different contexts. This kind of contextualised approach to teaching and learning is important for sustainability education, connecting the local and global, and acquiring a sense of *cynefin*.

In practice, UWTSD presents impressive numbers in learner engagement through entrepreneurship, as demonstrated by its HESA ranking and extensive alumni participation. Within the teacher training programmes delivered at UWTSD, all are mapped against the "Education for Sustainable Development" plan. The aim is for trainee teachers to "understand their professional responsibilities in relation to ESD, with particular regard to the development of practice and engagement within the classroom, and the ability to understand, critically evaluate and adopt thoughtful sustainability values" [51].

This is supported in UWTSD's approach to assessment, which encourages educators to become increasingly effective in their assessment approaches and methods in relation

to measuring—and fostering—entrepreneurial skills. This includes for learners to be increasingly successful to self-evaluate and identify their next steps in learning and more effective means of self-regulation. For example, UWTSD has been involved in a European project to gather, and make available in an accessible way, assessment methods, tools, and examples of increasingly differentiated entrepreneurial education assessment approaches for educators. Relatedly, there is a strong focus in UWTSD's own work on assessment for learning. The elective course "Enterprise Educators" is built around different rounds of formative feedback. Similarly, UWTSD learners participate in the assessment process, for example, through reflection. UWTSD has made the EU's EntreComp framework work for itself in this regard. Such assessment methods can enhance learner engagement and agency and support the cultivation of action competence in the spirit of sustainability education including the potentials for social change.

Furthermore, UWTSD is focussed on fostering entrepreneurial skills in their learners. Learning and teaching thus foster skills such as critical thinking, problem solving, and decision-making, as well as the ability to see how value of different kinds (financial, cultural, social, and learning) may be generated. Teacher trainees in particular are encouraged to put these skills into action and be experimental in their teaching. For over 10 years, UWTSD has been involved in different international (education) innovation projects focussed on mainstreaming entrepreneurial education among teachers. In a similar but wider manner, creativity and innovation are at the heart of UWTSD's mission to graduate employment. This allows all learners to become enterprising and be able to manage their and others' resources, giving them agency and skills to influence their social, economic, and environmental conditions, both locally and globally. UWTSD's Sustainability Statement focuses on providing meaningful education that considers social responsibility and the needs of future generations. The university aims to "utilise our collective skills, knowledge and technology to enable the University and its graduates to offer solutions to the most urgent societal challenges—in Wales and further afield. We are also committed to building a sustainable society driven through enterprising innovation and entrepreneurship" [54] (p. 3).

## 6. Conclusions

The findings are not generalisable and were not meant to be. They are limited to a case of how the work of UWTSD in the area of entrepreneurial education emerges as practice in the spirit of sustainable education and has the potential to support social change. It is a descriptive and analytical study where core elements in the progressive CFW are illustrated against the work of UWTSD to exemplify how entrepreneurial education can provide affordances to support education for social change.

A novelty of this research is that the aspects of entrepreneurial education are highlighted as a curriculum ideology (CFW) and as examples in practice at the university level in teacher education. Tables 4–6 illustrate the essence of the research and can be used to develop similar frameworks to scrutinise other curricula focusing on social change and entrepreneurial education. The presented examples are related to the professional context and practical implementation is illustrated plastically, and they show how entrepreneurial education approaches can embed sustainable thinking and education for social change. The findings can be informative and explanatory for teachers and teacher educators looking for ways to enhance sustainability education and social change.

Further research could deepen our findings and contextualise how a progressive curriculum can pan out in practice; for example, research on how entrepreneurial education emerges in school practice and other educational settings in Wales, looking for signs of the elements presented in the framework (Tables 4–6). Similar research in other countries focusing on how entrepreneurial education can support sustainability education and social change could also benefit from applying the framework or adjusting it accordingly.

**Author Contributions:** R.W. had a leading role in gathering data and designing the approach and S.R.J. handled the overall supervision. Both authors worked in tight collaboration on all other parts

of the research and writing up the article. Both authors have read and agreed to the published version of the manuscript.

**Funding:** This research was funded by the University of Iceland Research Fund.

**Institutional Review Board Statement:** Not applicable.

**Informed Consent Statement:** Not applicable.

**Data Availability Statement:** Most data is available online and cited in the reference list, except course handbooks from the UWTSD that we have not permission to share publicly.

**Acknowledgments:** We want to thank the University of Iceland Research Fund for the grant allocated to this research and the Educational Research Institute at the University of Iceland for advice and administrative support. The authors are grateful to Andy Penaluna, the editors of this issue, as well as anonymous referees for feedback on earlier versions of this paper. We also want to thank our language editor Jakob Maas for his excellent and swift services.

**Conflicts of Interest:** The authors declare no conflict of interest. The funders had no role in the design of the study; in the collection, analyses, or interpretation of data; in the writing of the manuscript, or in the decision to publish the results.

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
