# Peer review of "Education for Social Change: The Case of Teacher Education in Wales"

_sustainability, doi:10.3390/su13158574_

Round 1

Reviewer 1 Report

Congratulation for your extensive work about the education for social change in Wales educational system.

Author Response

Thank you for your supportive comments. See attached file with responses to your and the two other reviews.

Reviewer 2 Report

First of all, I would like to congratulate the authors on the work presented. The research develops an interesting study around Education for social change: the case of teacher education in 2 Wales

The authors indicate that the research paper does not follow a traditional structure, as their literature review is closely linked to the theoretical framework we use for analysis. Nevertheless,

it is suggested that the sections of the article be reordered in accordance with the Journal Article Reporting Standards for Qualitative Research. At least, it is necessary that the authors develop the section on Method in a more exhaustive manner.

Different points are proposed below for the authors that could be taken into account.

Method

Research design overview

  • Summarize the research design (data-collection strategies, data analytic strategies, and, if illuminating, approaches to inquiry (e.g., in many researches it could be useful descriptive, interpretive, feminist, psychoanalytic, postpositivist, critical, postmodern or constructivist, pragmatic approaches).
  • Provide the rationale for the design selected.

Although the study provides a method description that other investigators should be able to follow, it is not required that other investigators arrive at the same conclusions, but rather that their method should lead them to conclusions with a similar degree of methodological integrity.

Processes of qualitative research are often iterative versus linear, may evolve through the inquiry process, and may move between data collection and analysis in multiple formats. As a

result, data collection and analysis sections might be combined.

For the reasons above and because qualitative methods often are adapted and combined creatively, requiring detailed Description and rationale, an average qualitative Method section typically is longer than an average quantitative Method section

Study participants or data sources

Researcher description

  • Describe the researchers’ backgrounds in approaching the study, emphasizing their prior understandings of the phenomena under study (e.g., interviewers, analysts, or research team).
  • Describe how prior understandings of the phenomena under study were managed and/or influenced the research (e.g., enhancing, limiting, or structuring data collection and analysis).
  • Authors: Prior understandings relevant to the analysis could include, but are not limited to,

descriptions of researchers’ demographic/cultural characteristics, credentials, experience with phenomena, training, values, decisions in selecting archives or material to analyze.

Researchers differ in the extensiveness of reflexive self-description in reports.

Data sources

  • Provide the numbers of documents analyzed.
  • Describe the demographics/cultural information, or characteristics of data sources that might influence the data collected.
  • Describe existing data sources, if relevant (e.g., newspapers, Internet, archive).
  • Provide data repository information for openly shared data, if applicable.
  • Describe archival searches or process of locating data for analyses,

Document recruitment

Describe the recruitment process

  • Provide any changes in numbers through attrition and final number of sources (if relevant, refusal rates or reasons for dropout).
  • Describe the rationale for decision to halt data collection (e.g.,

saturation).

The order of the recruitment process and the selection process and their contents may be determined in relation to the authors’ methodological approach.

Data collection

Data collection/identification procedures

  • State the form of data collected (e.g., interviews, questionnaires, media, observation).
  • Describe the origins or evolution of the data-collection protocol.
  • Describe any alterations of data-collection strategy in response to the evolving findings or the study rationale.
  • Describe the data selection or collection process (e.g., were others present when data were collected, number of times data were collected, duration of collection, context).
  • Convey the extensiveness of engagement (e.g., depth of engagement, time intensiveness of data collection).

Recording and data

transformation

  • Identify data audio/visual recording methods, field notes, transcription processes used.

Analysis

Data-analytic strategies

  • Describe the methods and procedures used and for what purpose/goal.
  • Explicate in detail the process of analysis, including some discussion of the procedures (e.g., coding, thematic analysis, etc.) with a principle of transparency.
  • Describe coders or analysts and their training, if not already described in the researcher description section (e.g., coder selection, collaboration groups).
  • Identify whether coding categories emerged from the analyses or were developed a priori.
  • Identify units of analysis (e.g., entire transcript, unit, text) and how units were formed, if applicable.
  • Describe the process of arriving at an analytic scheme, if applicable (e.g., if one was developed before or during the analysis or was emergent throughout).
  • Provide illustrations and descriptions of their development, if relevant.
  • Indicate software, if used.

Descriptions should be provided, however, in accessible terms in relation to the readership.

Methodological integrity

Demonstrate that the claims made from the analysis are warranted and have produced findings with methodological integrity. The procedures that support methodological integrity (i.e., fidelity and utility) typically are described across the relevant sections of a paper, but they could be addressed in a separate section when elaboration or emphasis would be

helpful.

Research does not need to use all or any of the checks (as rigor is centrally based in the iterative process of qualitative analyses, which inherently includes checks within the evolving, self-correcting iterative analyses), but their use can augment a study’s methodological integrity. Approaches to inquiry have different traditions in terms of using checks and which checks are most valued.

Issues of methodological integrity could include:

  • Assess the adequacy of the data in terms of its ability to capture forms of diversity most relevant to the question, research goals, and inquirí approach.
  • Describe how the researchers’ perspectives were managed in both the data collection and analysis (e.g., to limit their effect on the data collection, to structure the analysis).
  • Demonstrate that findings are grounded in the evidence (e.g.,using quotes, excerpts, or descriptions of researchers’ engagement in data collection).
  • Demonstrate that the contributions are insightful and meaningful (e.g., in relation to the current literature and the study goal).
  • Present findings in a coherent manner that makes sense of contradictions or disconfirming evidence in the data (e.g., reconcile discrepancies, describe why a conflict might exist in

the findings).

Demonstrate consistency with regard to the analytic processes (e.g., analysts may use demonstrations of analyses to Support consistency, describe their development of a stable perspective, interrater reliability, consensus) or describe responses to inconsistencies, as relevant (e.g., coders switching mid-analysis, an interruption in the analytic process). If alterations in methodological integrity were made for ethical reasons, explicate

those reasons and the adjustments made.

  • Describe how support for claims was supplemented by any checks added to the qualitative analysis.

 Examples of supplemental checks that can strengthen the research may include:

  • Transcripts/data collected returned to participants for feedback.
  • Triangulation across multiple sources of information, findings, or
  •  
  • Data displays/matrices.
  • In-depth thick description, case examples, illustrations.
  • Structured methods of researcher reflexivity (e.g., sending
  • memos, field notes, diary, log books, journals, bracketing).
  • Checks on the utility of findings in responding to the study
  • problem (e.g., an evaluation of whether a solution worked).

It is recommended to include a section on conclusions, as well as to develop the limitation and perspectives of the study.

Author Response

Thank you for your constructive and helpful comments. See attached file with responses to your and the two other reviews.

Reviewer 3 Report

The case study structure is undoubtedly not traditional, but the authors interpret the purpose of the study. The main level of the authors' analysis is the curriculum construction, which after 2006 significantly renewed the content analysis of changes in entrepreneurial education in its content and procedures. New curriculum developments have given noticeably greater scope for creativity and innovation, critical thinking and sustainable development in Welsh education. Following the relatively laconic first and second chapters, the authors describe the curriculum development process in detail in Chapter 3, which presents the theoretical background. All this undoubtedly makes a significant contribution to the understanding of contemporary Welsh education from a comparative analytical point of view, but at the same time makes the whole study somewhat structurally disproportionate. An essential novelty of the authors' research is that the aspects of entrepreneurial education are highlighted. The 4-6 Tables are especially very useful constructs that perfectly illustrate the essence of the research. The presented examples are well related to the professional context, the practical implementation is illustrated plastically. The following Chapter 5 summarizes the main findings of the discussion in terms of sustainability thinking and actions for social change. The authors articulate their conclusions and predictions for the future that encourage discussion and further research, which are closely related to the progressive interpretation of sustainability thinking and actions for social change.

Author Response

(The authors gave the same response as above.)

Round 2

Reviewer 2 Report

I would like to thank the authors for this new version. Now I suggest to accept this contribution.